# Long-Chain Saturated Fatty Acids, Palmitic and Stearic Acids, Enhance the Repair of Photosystem II

**DOI:** 10.3390/ijms21207509

**Published:** 2020-10-12

**Authors:** Haruhiko Jimbo, Kensuke Takagi, Takashi Hirashima, Yoshitaka Nishiyama, Hajime Wada

**Affiliations:** 1Graduate School of Arts and Sciences, University of Tokyo, Komaba, Meguro-ku, Tokyo 153-8902, Japan; takagi-kensuke543@g.ecc.u-tokyo.ac.jp (K.T.); thirashima@bio.c.u-tokyo.ac.jp (T.H.); hwada@bio.c.u-tokyo.ac.jp (H.W.); 2Department of Biochemistry and Molecular Biology, Graduate School of Science and Engineering, Saitama University, Shimo-Okubo, Sakura-ku, Saitama 338-8570, Japan; nishiyama@mail.saitama-u.ac.jp

**Keywords:** free fatty acids, biodiesel, photoinhibition of PSII, cyanobacteria

## Abstract

Free fatty acids (FFA) generated in cyanobacterial cells can be utilized for the biodiesel that is required for our sustainable future. The combination of FFA and strong light induces severe photoinhibition of photosystem II (PSII), which suppresses the production of FFA in cyanobacterial cells. In the present study, we examined the effects of exogenously added FFA on the photoinhibition of PSII in *Synechocystis* sp. PCC 6803. The addition of lauric acid (12:0) to cells accelerated the photoinhibition of PSII by inhibiting the repair of PSII and the de novo synthesis of D1. α-Linolenic acid (18:3) affected both the repair of and photodamage to PSII. Surprisingly, palmitic (16:0) and stearic acids (18:0) enhanced the repair of PSII by accelerating the de novo synthesis of D1 with the mitigation of the photoinhibition of PSII. Our results show chemical potential of FFA in the regulation of PSII without genetic manipulation.

## 1. Introduction

Our sustainable future depends on new sources of renewable energy instead of limited petroleum. Free fatty acids (FFA), materials for biodiesel, are a new and prominent power resource. Microalgae, including cyanobacteria are suitable platforms for the production of FFA. Disruption of acyl-ACP synthetase (AAS) gene and expression of thioesterase (TE) gene are the most conventional methods for improving FFA production in microalgal cells [1]. In addition, the expression of an RND-type exporter for FFA in *Synechococcus elongatus* PCC 7942 (hereafter *Synechococcus*) raises the rate of FFA production up to 2 mg L^−1^ h^−1^ [2]. However, accumulated FFA severely inhibit the photosynthetic activity [3,4]. To achieve further production of FFA in cyanobacteria, it is essential to overcome the FFA-induced inhibition of photosynthesis.

Light is a driving force for photosynthesis; however, excessive light is harmful for photosynthesis. Photosystem II (PSII) is particularly sensitive to light, and the inactivation by light (photodamage) occurs under any intensity of light [5]. The reduction in PSII activity by excess light is referred to as the photoinhibition of PSII [6]. Damaged PSII is rapidly repaired by a proteolysis-based repair cycle. Photoinhibition becomes apparent when the rate of photodamage to PSII exceeds the rate of the repair of damaged PSII [7]. The repair of damaged PSII depends on the de novo synthesis of D1, a reaction center of PSII [8]. Thus, the rate of photodamage under strong light can be monitored in the presence of lincomycin or chloramphenicol, inhibitors of protein synthesis [9]. Photoinhibition of PSII is accelerated by the combination of strong light and excess amounts of FFA [10]. In an *aas* mutant of *Synechococcus*, increased amounts of FFA inhibit cellular growth and accelerate the photoinhibition of PSII by accelerating the rate of photodamage to PSII [11]. Among FFA molecules, linoleic (18:2) and linolenic acids (18:3) severely inhibit photosynthetic electron transport and the cellular growth of cyanobacteria [4,12]. Further investigation of the effect of FFA on cellular growth reports that the sensitivity of cyanobacteria to extracellular FFA is improved in the *aas* mutant [12,13]. These results suggest that inhibitory effects of FFA on the cellular growth and photosynthesis occur when FFA are activated as acyl-ACP. The action of FFA has been further investigated in respiratory electron transport. FFA have a protonophoric effect and inhibit the formation of proton motive force across the membrane, which enhances the production of reactive oxygen species (ROS) [14]. However, the molecular specificity and the inhibitory mechanisms of FFA in the photoinhibition of PSII remain unclear.

In the present study, *Synechocystis* sp. PCC 6803 (hereafter *Synechocystis*) was incubated in the presence of various FFA molecules, and we examined the effects of exogeneous FFA on the photoinhibition of PSII. Lauric acid (12:0) suppressed the repair of PSII by inhibiting the de novo synthesis of D1, with the resultant acceleration of photoinhibition of PSII. α-linolenic acid (18:3) accelerated photodamage to PSII. On the other hand, palmitic (16:0) and stearic acids (18:0) alleviated the photoinhibition of PSII by enhancing the repair of PSII via the accelerating synthesis of D1. From these observations, we report here the FFA-related photoinhibition by affecting the de novo synthesis of D1.

## 2. Results

### 2.1. Effects of Exogenous FFA on the Photoinhibition of PSII Under Strong Light

We examined the effects of exogenously added FFA on the activity of PSII prior to photoinhibition assay. The activities of PSII were 774 ± 43, 823 ± 74, 705 ± 25, 814 ± 121, 772 ± 124, and 726 ± 94 μmol O_2_ mg^−1^ Chl h^−1^ in the presence of 12:0, 14:0, 16:0, 18:0, 18:1, and 18:3, respectively. Since the PSII activity in the absence of FFA was 851 ± 45 μmol O_2_ mg^−1^ Chl h^−1^, the exogeneous FFA did not significantly affect the activity of PSII. It should be noted that exogeneous FFA are immediately absorbed by *Synechocystis* cells [12], and the consequent effects of FFA become apparent on PSII activity at least within a few minutes. The photoinhibition of PSII results in the inhibition of cell growth in plants, algae, and cyanobacteria [15,16]. In *Synechocystis*, 12:0 and 18:3 inhibit cell growth, and 16:0 and 18:0 accelerate it [4,13]. To investigate the effects of FFA on the photoinhibition of PSII, we exposed *Synechocystis* cells to strong light at 1500 μmol photons m^−2^ s^−1^ at 32 °C in the absence or presence of various FFA at the concentration of 0.1 mM. The activity of PSII in the absence of FFA decreased to 25% of the initial level in 80 min under strong light (Figure 1). The activity of PSII in the presence of 18:1 decreased at similar rates to those in the absence of FFA (Figure 1b). However, the addition of either 16:0 or 18:0 alleviated the photoinhibition of PSII: the activity of PSII retained at 45% of the initial level in 80 min (Figure 1a,b). By contrast, the addition of 12:0 and 18:3 accelerated the photoinhibition of PSII: the activity of PSII was less than 15% of the initial level in 80 min (Figure 1a,b). It is also reported that the addition of 18:3 inhibits the rate of photosynthetic electron transport under weak light [12]. The PSII activity in the presence of 14:0 fell slightly faster than that in the absence of FFA (Figure 1a,c), indicating 14:0 had a more milder effect on the photoinhibition of PSII than 12:0 or 18:3. By the comparison of 18:0, 18:1 and 18:3 accelerated the photoinhibition of PSII (Figure 1b,c). These observations suggest that FFA that are shorter than 14:0 or more unsaturated accelerate the photoinhibition of PSII to a greater degree. To test the effects of FFA on the PSII activity further, we monitored the changes in PSII activity in the dark in the presence of various FFA for 80 min. Since PSII repair requires light, the change in PSII activity in the dark indicates the thermostability of PSII. Incubation in the absence or presence of FFA under the dark did not significantly affect PSII activity (Figure 2), indicating that the addition of various FFA has no effects on the thermostability of PSII. These results show that FFA-related photoinhibition depends on both molecular species of FFA and the light.

### 2.2. Effects of Exogenous FFA on Photodamage to PSII under Strong Light

Accumulated FFA in an *aas* mutant of *Synechococcus* increase the rate of photodamage to PSII [11]. Here, we investigated the effects of various FFA on the photodamage to PSII. Photodamage to PSII was analyzed in the presence of lincomycin, an inhibitor of protein synthesis. The activity of PSII in the presence of 12:0, 14:0, 16:0, 18:0, and 18:1 decreased at rates similar to those in the absence of FFA (Figure 3), indicating these FFA species have no effects on photodamage to PSII. Thus, it appears that the repair of PSII might be inhibited in the presence of 12:0 and 14:0 and enhanced in the presence of 16:0 and 18:0 (Figure 1 and Figure 3). Photodamage to PSII was accelerated in the presence of 18:3 (Figure 3b), indicating that extended photoinhibition of PSII in the presence of 18:3 was due to the both the inhibition of repair of PSII and increased sensitivity of PSII to photodamage. Thus, saturated long-chained FFA might protect the repair of PSII, whereas polyunsaturated FFA might accelerate photodamage to PSII.

### 2.3. Effects of FFA on the De Novo Synthesis of D1 under Strong Light

The de novo synthesis of D1 plays a central role in the repair of PSII [17]. To examine the effects of 12:0, 16:0 and 18:0 on the de novo synthesis of D1 under strong light, newly synthesized proteins in thylakoid membranes were labeled with ^35^S-labeled Met and Cys in the presence of various FFA under strong light at 1,500 μmol photons m^−2^ s^−1^ at 32 °C. Figure 4a shows a time course of radiogram of labeled proteins in thylakoid membranes, which were separated by SDS-PAGE. In cells, in the absence of FFA, mature D1 (D1) polypeptides were detected as a single band (Figure 4a). However, in all cells in the presence of FFA, premature (pD1) and processing intermediate D1 (iD1) polypeptides were detected as well (Figure 4a). In cells, in the presence of 12:0 or 18:0, iD1 accumulated at higher levels than pD1 or D1 (Figure 4a). In all cells, with or without FFA, the level of labeled D1 increased under strong light (Figure 4b). The rate of synthesis of D1 in cells with 12:0 decreased to 67% of that in cells without FFA (Figure 4b). Therefore, 12:0 inhibited repair of damaged PSII by suppressing the maturation of D1 under strong light. The inhibition of protein synthesis by 12:0 was observed in almost all proteins in thylakoid membranes (Figure 4a), suggesting that 12:0 adversely affects whole protein synthesis. The rates of de novo synthesis of D1 in cells in the presence of 16:0 and 18:0 were 22 and 30% higher than that in cells in the absence of FFA, respectively (Figure 4b). The stimulation was observed only in D1 (Figure 4a). This result indicates that FFA might affect both de novo synthesis and maturation of D1 under strong light.

## 3. Discussion

### 3.1. Acceleration of Photoinhibition of PSII by Medium-Chain Fatty Acids and Polyunsaturated Fatty Acids

In the present study, we demonstrated that medium-chain fatty acids (MCFA) such as 12:0 accelerated the photoinhibition of PSII by suppressing the de novo synthesis of thylakoid proteins including D1 (Figure 2 and Figure 4). Fatty acids bound to phosphatidylglycerol (PG), which is the only phospholipid in cyanobacterial membrane lipids, are rapidly cleaved off by lipases and the resultant lyso-PGs are reacylated; namely, remodeling of PG takes place in the cells [18], suggesting that exogeneous FFA incorporated into the cells can be bound to PG molecules via the activation of FFA to acyl-ACP by AAS. PG molecule anchors D1 polypeptides and enhance PSII repair [19,20]. Since the mutant of AAS tolerated to 12:0 [13], the incorporation of MCFA into PG molecules might inhibit the de novo synthesis of thylakoid proteins that is required for the repair of PSII.

Polyunsaturated fatty acids (PUFA), such as 18:3, accelerated photodamage to PSII (Figure 3). The accelerated photodamage by 18:3 was observed only under light (Figure 2), indicating the action of 18:3 on photodamage depends on photosynthetic electron transport. The mutant of AAS was tolerant to 18:3 [12,13], suggesting that effect of 18:3 on the cyanobacteria appears after the activation of 18:3 to 18:3-ACP by AAS. Unsaturation of lipids increases membrane fluidity and permeability, which might inhibit the formation of proton motive force across the membrane. Uncoupling ΔpH across the thylakoid membrane by the addition of NH_4_Cl or nigericin extends photodamage to PSII [21,22], which implies that incorporation of 18:3 into membrane lipids inhibits the formation of thylakoid ΔpH as well, which extends photodamage of PSII. In addition, PUFA easily become targets of ROS and the peroxidation of PUFA induces the photoinhibition of PSII [23,24]. A lack of α-tocopherol, a scavenger of singlet oxygen, increases lipid peroxidation and inhibits PSII repair under strong light [25,26]. Therefore, it is likely that the peroxidation of exogeneous PUFA inhibits repair of PSII as well.

### 3.2. Alleviation of Photoinhibition of PSII by Saturated Long Chain Fatty Acids

The supplementation of 16:0 and 18:0 had positive effects on the photoinhibition of PSII (Figure 1). *Synechocystis* does not have the complete metabolic cycle for the degradation of fatty acids, such as β-oxidation [27], indicating that exogeneous FFA might not be metabolized or used for the substrates for the respiration in the cells. In the cycle of fatty acid synthesis, 16:0 is the substrate for elongation to 18:0. The effect of 16:0 in photoinhibition was similar to that of 18:0. Although the molecular mechanisms for the contribution of FFA to repair of PSII have remained unclear, both 16:0 and 18:0 accelerated the de novo synthesis of D1 under strong light (Figure 4). In the cells with 16:0 or 18:0, iD was detected. However, pD1 was accumulated in the cells incubated with 18:0 as well as iD1. pD1 is processed by CtpA, a protease that generates intermediate D1 polypeptides [28]. Our observations suggest that saturated FFA might affect the maturation of D1. Chlorophylls are loaded into pD1 followed by the C-terminal processing by CtpA [29]. Thus, the increased amount of iD1 in cells incubated with 16:0 and 18:0 might be due to the faster loading of chlorophyll *a* into pD1 during PSII repair. Biosynthesis of fatty acids is a rate-limiting step in lipid biosynthesis. The exogenously added 16:0 and 18:0 are incorporated into the cells and converted to 16:0-ACP and 18:0-ACP, respectively, by AAS, which are the substrates for lipid biosynthesis, and thus facilitate lipid biosynthesis. PG and DGDG are required for the construction of OEC, which contributes rapid repairing of PSII [30,31], and PG enhances the de novo synthesis of D1 by anchoring D1 peptides into the membrane [19,20]. Therefore, exogeneous 16:0 or 18:0 might increase the biosynthesis of DGDG and PG, which both enhance PSII repair. In cyanobacteria, saturated fatty acids synthesized by fatty acid synthase are utilized for lipid biosynthesis and then desaturated to unsaturated fatty acids by acyl-lipid desaturases, which introduce double bonds into fatty acids bound to lipids [32]. Thus, exogenous saturated fatty acids can be utilized for lipid biosynthesis after activation to acyl-ACPs by AAS. By contrast, exogenous unsaturated fatty acids are incorporated into lipids by the exchange of acyl groups of lipids, deacylation of lipids by lipases and reacylation of lyso-lipids with acyl-ACPs by acyltransferases, and do not facilitate lipid biosynthesis. The different effects between saturated and unsaturated fatty acids on photoinhibition might be caused by such different utilizations for lipid biosynthesis.

### 3.3. Future Perspectives

In the present study, we showed the diverse effects of FFA species on the photoinhibition of PSII. Our findings show a potential chemical regulation of photosynthesis with FFA or FFA-derived molecules. Numerous studies use genetically modified organisms (GMO) to improve the ability of photoprotection and the biomass yield [33,34,35]. However, GMO are not fully accepted by society, and require extra effort to apply to industries. Our findings have a big advantage in regulating photosynthetic activity under strong light without any genetic modifications. In addition, FFA are abundant carbon resources in the earth and are rapidly turned over by cellular metabolisms in microorganisms. These natural characteristics of FFA provide us with a new tool to control photosynthetic activity in nature with the lighter environmental load than artificial chemical compounds.

## 4. Materials and Methods

### 4.1. Strains and Culture Conditions

Cells of a glucose-tolerant strain of *Synechocystis* sp. PCC 6803 were grown photoautotrophically at 32 °C in liquid BG11 medium under light at 70 μmol photons m^−2^ s^−1^, with aeration by sterile air, as described previously [36]. Cells in cultures with an optical density at 730 nm of 1.0 ± 0.1 (equal to about 3.6 μg mL^−1^ chlorophyll *a*) were used for assays. White halogen light was used as the light source in all experiments.

### 4.2. Photoinhibition of PSII

Cell cultures in the presence of 0.1 mM FFA were exposed to light at 1500 μmol photons m^−2^ s^−1^ at 32 °C for designated periods of time to induce the photoinhibition of PSII, as described previously [36]. For assays of photodamage, 200 μg mL^−1^ lincomycin was added to the suspension of cells just before the onset of illumination. Cell cultures were also incubated in the dark at 32 °C for designated periods of time. The activity of PSII under 1500 μmol photons m^−2^ s^−1^ was measured at 32 °C in terms of the evolution of oxygen in the presence of 1 mM 1,4-benzoquinone and 1 mM K_3_Fe(CN)_6_ with a Clark-type oxygen electrode (Hansatech Instruments, King’s Lynn, UK). FFA compounds are purchased from FujiFilm Wako (Tokyo, Japan).

### 4.3. Labeling of Proteins Synthsized De Novo

Prior to the labeling experiments, cell cultures were incubated at 32 °C under strong light at 1,500 μmol photons m^−2^ s^−1^ in the presence of 0.1 mM FFA for 40 min. Then, pulse labeling of proteins was started by the addition of 5 μCi mL^−1 35^S-labeled methionine and cysteine (Easy Tag EXPRE^35^S; PerkinElmer). A total of 6 mL of culture was taken at the designed times, and the reaction was terminated by the addition of 2 mM cold Met and Cys each. Proteins in thylakoid membrane were isolated by breaking cells with beads-beater followed by the centrifugation, and proteins (equal to 3.2 μg Chl) were analyzed by 12.5% SDS-PAGE containing 6 M urea, as described previously [37]. Labeled proteins were visualized by autoradiograms, and the levels of labeled D1 were quantified by using Image J2.

## Figures and Tables

**Figure 1 ijms-21-07509-f001:**
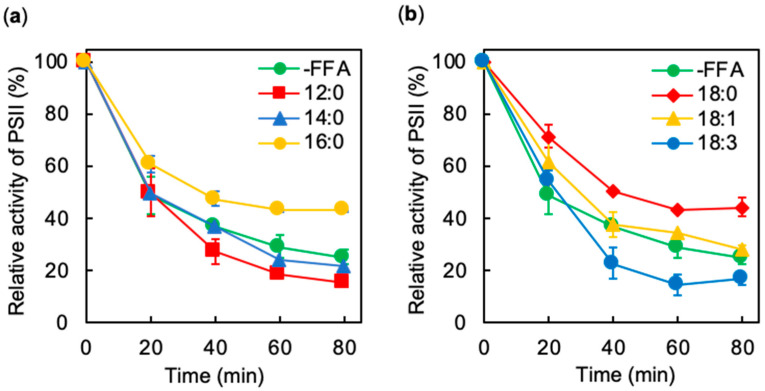
Photoinhibition of PSII in *Synechocystis* cells under strong light in the presence of various free fatty acids (FFA). (**a**) *Synechocystis* cells were incubated at 32 °C under strong light at 1500 μmol photons m^−2^ s^−1^, with ambient aeration, in the absence (-FFA); green circles or presence of 0.1 mM lauric acid (12:0); red squares, myristic acid (14:0); blue triangles, or palmitic acid (16:0); yellow circles. (**b**) Same condition as (a) in the absence (-FFA); green circles or presence of 0.1 mM stearic acid (18:0); red diamonds, oleic acid (18:1); yellow triangles, and α-linolenic acid (18:3); blue circles. The activity of PSII of the cells was measured at 32 °C in terms of the evolution of oxygen in the presence of 1 mM 1,4-benzoquinone and 1 mM K_3_Fe(CN)_6_. Values are means ± S.D. (bars) of results from three independent experiments.

**Figure 2 ijms-21-07509-f002:**
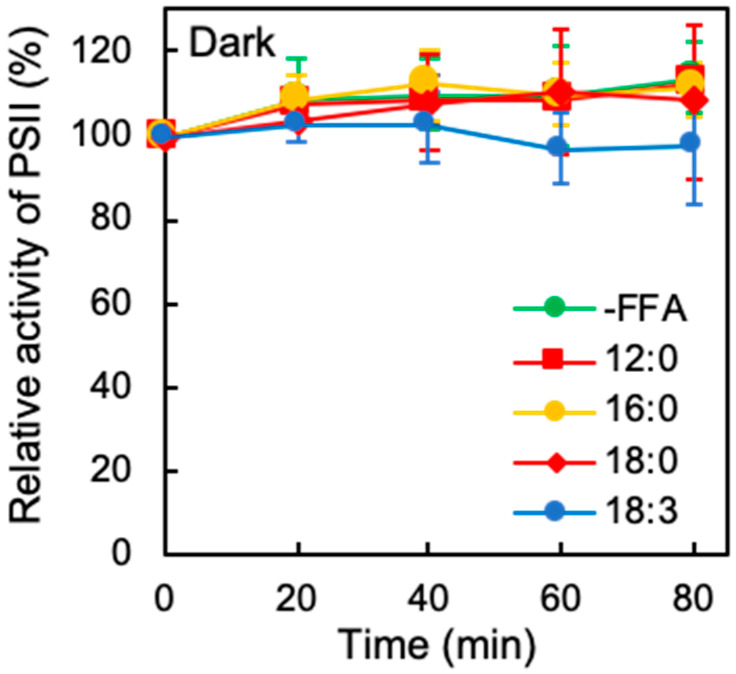
Effect of FFA on PSII activity in the dark. *Synechocystis* cells were incubated at 32 °C in the dark, with ambient aeration, in the absence (-FFA); green circles or presence of 0.1 mM lauric acid (12:0); red squares, palmitic acid (16:0); yellow circles, stearic acid (18:0); red diamonds, and α-linolenic acid (18:3); blue circles. The activity of PSII of the cells was measured at 32 °C in terms of the evolution of oxygen in the presence of 1 mM 1,4-benzoquinone and 1 mM K_3_Fe(CN)_6_. Values are means ± S.D. (bars) of results from three independent experiments.

**Figure 3 ijms-21-07509-f003:**
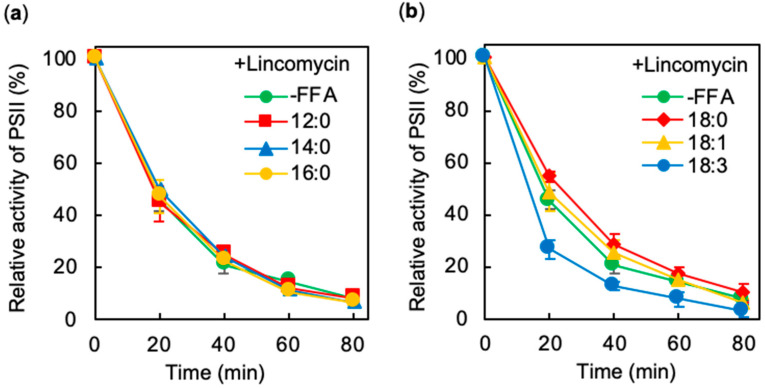
Photodamage to PSII in *Synechocystis* cells under strong light in the presence of 0.2 mg mL^−1^ lincomycin and various FFA. (**a**) *Synechocystis* cells were incubated at 32 °C under strong light at 1,500 μmol photons m^−2^ s^−1^, with ambient aeration, in the absence (-FFA) or presence of 0.1 mM lauric acid (12:0; red squares), myristic acid (14:0; blue triangles), and palmitic acid (16:0; yellow circles). (**b**) Same condition as (a) in the absence (-FFA; green circles) or presence of 0.1 mM stearic acid (18:0; red diamonds), oleic acid (18:1; yellow triangles), and α-linolenic acid (18:3; blue circles). The activity of PSII of the cells was measured at 32 °C in terms of the evolution of oxygen in the presence of 1 mM 1,4-benzoquinone and 1 mM K_3_Fe(CN)_6_. Values are means ± S.D. (bars) of results from three independent experiments.

**Figure 4 ijms-21-07509-f004:**
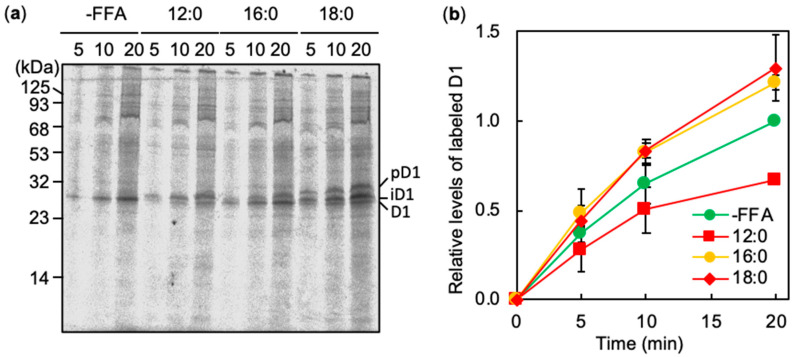
Effects of FFA on the de novo synthesis of D1 under strong light. Proteins in *Synechocystis* cells in the presence of 0.1 mM FFA were pulse- labeled by incubation of cells at 32 °C, for the indicated times, under strong light at 1500 μmol photons m^−2^ s^−1^ with standard aeration, in the presence of ^35^S-labeled Met and Cys at the concentration of 5 μCi mL^−1^. Thylakoid membranes were isolated, and proteins were separated by SDS-PAGE containing 6 M urea. (**a**) Representative radiogram of pulse-labeled proteins from thylakoid membranes. (**b**) Quantitation of the relative levels of labeled D1. Values are means ± S.D. (bars) of results from three independent experiments. Three bands of D1 polypeptides are indicated as pD1; premature D1, iD1; intermediate D1, and D1; matured D1, respectively. The level of labeled D1 in the absence of FFA for 20 min was taken as 1.0.

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
