# Peer review of "Long-Chain Saturated Fatty Acids, Palmitic and Stearic Acids, Enhance the Repair of Photosystem II"

_ijms, 2020, doi:10.3390/ijms21207509_

Round 1
Reviewer 1 Report
The manuscript titled „Long-chain saturated fatty acids, palmitic and stearic acids, enhance the repair of photosystem II” provides new data about the effects of exogenously added free fatty acids (FFA) on the photoinhibition of PSII in Synechocystis sp. PCC 6803. It would be important information if we plan to use cyanobacteria for biofuel production, because it was shown earlier that some kind of accumulated FFA severely inhibit photosynthetic activity and the growth of cyanobacteria. Authors discussed that according to their measurements, while 12:0 and 18:3 FFAs accelerated photoinhibition of PSII, 16:0 and 18:0 FFAs had positive effect on photoinhibition by increasing the de novo synthesis of D1 under strong light. Although, the molecular mechanism of this effect remained unclear.
Comments
- I missed a short summary of the studied FFAs in the Methods (source, concentration).
- Description of the measurement of PSII activity in the dark is missing in the Methods and the explanation of this measurement in the Result is ambiguous.
- Why did not authors measure the effect of 18:3 FFA on the de novo synthesis of D1 protein? I would be interesting to see its effect also on the repair of PSII.
- Discussion part of the paper is rather speculative. For example at line 163 „ De novo synthesis of D1 requires PG [19]”. According to ref .19 only the dimerization of PSII core monomers requires PG and the de novo synthesis of D1 does not. At line 168: „excess amount of 12:0 might induce the production of ROS”. There is no experimental or literatural indication on the effect of 12:0 on the ROS production in this manuscript.
Specific comments
- Controversial statement at line 73-74: „The activity of PSII in the absence or presence of 18:1 decreased at similar rates” and at line 81: 18:1 and … accelerated the photoinhibition of PSII”
- Controversial or not obvious phrasing in the line 112: „the repair of PSII might be inhibited in the presence of 12:0” and line 115-116: „saturated FFA might protect the repair of PSII”
Author Response
We appreciate the Reviewer 1 for critical comments provided on the previous version of our manuscript. The comments were very helpful for improving our manuscript. We have taken these comments into account and revised the manuscript.
Comments
- I missed a short summary of the studied FFAs in the Methods (source, concentration).
- Description of the measurement of PSII activity in the dark is missing in the Methods and the explanation of this measurement in the Result is ambiguous.
----Response to the comments 1 and 2----
We completely agree about fixing the missing parts about the source and concentration of FFA and description of PSII activity in the cells with FFA incubated in the dark. We added the information in Materials and Methods (Lines 291-298 in the revised text). - Why did not authors measure the effect of 18:3 FFA on the de novo synthesis of D1 protein? I would be interesting to see its effect also on the repair of PSII.
----Response to the comment 3----
We are also very interested in the effects of 18:3 on the de novo synthesis of D1 protein. But the mechanisms involved might be more complicated than those of other FFA because 18:3 affects not only repair, but also photodamage to PSII. Thus, in the present study, We chose 12:0, 16:0, and 18:0 that affect only the repair of PSII to make the story simple and to avoid the confusion to the readers. We are planning to investigate the mechanisms how 18:3 affects the both repair and photodamage to PSII in the future. - Discussion part of the paper is rather speculative. For example at line 163, De novo synthesis of D1 requires PG [19]”. According to ref .19 only the dimerization of PSII core monomers requires PG and the de novo synthesis of D1 does not. At line 168: „excess amount of 12:0 might induce the production of ROS”. There is no experimental or literatural indication on the effect of 12:0 on the ROS production in this manuscript.
----Response to the comment 4----
We agree with your points. We edited line 167 and deleted the part of ROS production by 12:0 in the revised text.
Specific comments
- Controversial statement at line 73-74: „The activity of PSII in the absence or presence of 18:1 decreased at similar rates” and at line 81: 18:1 and … accelerated the photoinhibition of PSII”
----Response to the specific comment 1----
In lines 73-74, the activities of PSII in the absence and presence of 18:1 were compared. In line 81, the effect of 18:1 on PSII activity was compared with that of 18:0. Since 18:0 alleviated photoinhibition of PSII, mono-unsaturation seems accelerated it. It might be complex to be explained, so we edited line 74 and 81 in the revised text. - Controversial or not obvious phrasing in the line 112: „the repair of PSII might be inhibited in the presence of 12:0” and line 115-116: „saturated FFA might protect the repair of PSII”
----Response to the specific comment 2----
In lines 115-116, the phrase "saturated FFA" means 16:0 and 18:0. We changed “saturated FFA” to "saturated long-chain FFA" in the revised manuscript.
Reviewer 2 Report
The paper entitled “Long-chains saturated fatty acids, Palmitic and Stearic acids, enhance the repair of Photosystem II” by Jimbo and co-workers (IJMS-933050) describes and analysis of the effect of several free fatty acids (FFA), different by chain length and saturation levels, exogenously added of the cyanobacterium Synechocystis sp. PCC6803 on photosynthetic oxygen evolution, light-induced damage at the level of Photosystem II (PSII) and turn-over of PSII protein subunits.
It is shown that FFA can have contrasting effect; whereas most of the FFA tested lead to an increase in the light sensitivity of Photosystem II, two molecules, palmitic and stearic acid, appeared to reduce light-induced damage to PSII. The effect of these molecule does not appear to be a direct one, i.e. a direct prevention of light-induced damage, rather an indirect effect, promoting an increased turnover of PSII reaction centre subunit as evidenced in radioactive labelling de-novo synthesis experiments.
These represent an interesting new finding, which besides the specific effect of palmitic and stearic acid highlights the complexity of investigating photoinhibition (and photoprotetion) in vivo, when several processes contributes what is observed at cellular level.
The paper is generally well written and the results are clearly presented and interesting and the conclusion proposed by the authors are generally sound. Therefore I would recommend the paper for publication after minor revision.
The only issue which I find not fully substantiated is the discussion of the mechanism of increased inhibition, from page 6 line 188, until the end of paragraph 3.1. Differently from the rest of the paper, this section appears rather confusing and I found rather cumbersome to follow the author’s argumentation. This might be due to the fact they are trying to rationalise their finding on the basis of light-induced damage being ultimately initiated by malfunction of PSII donor side (OEC). Although this is a definite possibility, as discussed in the text, especially in visible light primary damage might also occur at the acceptor side. Henceforth, the interpretation of the results in terms of direct destabilization of the OEC by FFA (or FFA-ACP derivative) is way too speculative. One could for instance argue, equally arbitrarily, that changes in the fluidity/viscosity of the membrane might impact on PQ (or other electron carriers) diffusion, in turn limiting the acceptor side of PSII instead.
In the absence of direct evidence of OEC impairment, and, later one, for an increase/decrease of lipid peroxidation, I would recommend not to discuss these issues, or to much simplify the discussion restraining for proposing molecular/chemical mechanism.
Author Response
The paper entitled “Long-chains saturated fatty acids, Palmitic and Stearic acids, enhance the repair of Photosystem II” by Jimbo and co-workers (IJMS-933050) describes and analysis of the effect of several free fatty acids (FFA), different by chain length and saturation levels, exogenously added of the cyanobacterium Synechocystis sp. PCC6803 on photosynthetic oxygen evolution, light-induced damage at the level of Photosystem II (PSII) and turn-over of PSII protein subunits.
It is shown that FFA can have contrasting effect; whereas most of the FFA tested lead to an increase in the light sensitivity of Photosystem II, two molecules, palmitic and stearic acid, appeared to reduce light-induced damage to PSII. The effect of these molecule does not appear to be a direct one, i.e. a direct prevention of light-induced damage, rather an indirect effect, promoting an increased turnover of PSII reaction centre subunit as evidenced in radioactive labelling de-novo synthesis experiments.
These represent an interesting new finding, which besides the specific effect of palmitic and stearic acid highlights the complexity of investigating photoinhibition (and photoprotetion) in vivo, when several processes contribute what is observed at cellular level.
The paper is generally well written and the results are clearly presented and interesting and the conclusion proposed by the authors are generally sound. Therefore, I would recommend the paper for publication after minor revision.
The only issue which I find not fully substantiated is the discussion of the mechanism of increased inhibition, from page 6 line 188, until the end of paragraph 3.1. Differently from the rest of the paper, this section appears rather confusing and I found rather cumbersome to follow the author’s argumentation. This might be due to the fact they are trying to rationalise their finding on the basis of light-induced damage being ultimately initiated by malfunction of PSII donor side (OEC). Although this is a definite possibility, as discussed in the text, especially in visible light primary damage might also occur at the acceptor side. Henceforth, the interpretation of the results in terms of direct destabilization of the OEC by FFA (or FFA-ACP derivative) is way too speculative. One could for instance argue, equally arbitrarily, that changes in the fluidity/viscosity of the membrane might impact on PQ (or other electron carriers) diffusion, in turn limiting the acceptor side of PSII instead.
In the absence of direct evidence of OEC impairment, and, later one, for an increase/decrease of lipid peroxidation, I would recommend not to discuss these issues, or to much simplify the discussion restraining for proposing molecular/chemical mechanism
------------------------------------------------------------------------------------------------------
We appreciate the Reviewer 2 for the critical comments and suggestions.
We understand your comment that the discussion part is a bit hard to follow and contains too much speculations. Since the molecular mechanism of photodamage to PSII is not fully clarified, we deleted the part of the mechanism of photodamage and added biophysical aspects of 18:3 incorporated into membrane lipids in the revised manuscript (lines 174-178). We also simplified the possible action of lipid peroxidation in PSII repair (lines 179-181).
Reviewer 3 Report
The subject of the manuscript seems to be very intertested in reader who works in same fileds.
I suggest to accept this article without major reivse but, could you explain how you isolate thylakoid membrane in more detail?
Author Response
The subject of the manuscript seems to be very intertested in reader who works in same fileds.
I suggest to accept this article without major reivse but, could you explain how you isolate thylakoid membrane in more detail?
------------------------------------------------------------------------------------------------------
We appreciate the Reviewer 3 for understanding our work well and giving us a very positive comment. Concurring with the comment, we added the methods for isolation of thylakoid membrane in line 304 in the revised manuscript.
Reviewer 4 Report
This paper addresses the effect of Free Fatty Acids (FFA) on PSII repair cycle in the cyanobacteria Synechocystis PCC 6803. The medium-chain FFA Lauric acid (12:0) is proposed to inhibit PSII repair. The polyunsaturated α-linolenic acid (18:3) is proposed to inhibit PSII repair and also to enhance PSII damages. Palmitic (16:0) and steric (18:0) acids are shown as enhancing D1 synthesis, thus promoting PSII repair. The paper is clearly written and the experiments well interpreted. However, I feel that overall this paper would require additional experiments to support the conclusion drawn.
The authors proposed to use FFA to regulate photosynthetic activity (line 230) with the applied aim of improving the ability of photoprotection and biomass yield (line 227-228). Such as, the paper does not present any relevant data toward this aim. Therefore I would recommend performing a growth curve in presence of the different FFA (added once or at several times when the FFA turned-over is fast (as written line 231)).
In cyanobacteria, photosynthesis and respiration occur in the same membrane. Lines 164-170, the authors stress out that in animals FFA enhanced ROS production by inhibiting the respiratory electron transport. This raises the question of whether this is not happening as well in cyanobacteria? Hindering the photosynthetic linear electron transport chain would result in a more reduced plastoquinone pool. Consequently, PSII damages will be enhanced. This could explain the additional photodamages observed in presence of α-linolenic acid (18:3). To rule out this possibility, I would suggest experiments such as P700 re-reduction kinetics.
The oxygen evolution in the light is clearly impaired as shown in Figure 1. Since photosynthesis and respiration occur in the same membrane, both could be impaired. I would recommend verifying that the oxygen consumption in the dark is not changed in the presence of FFA.
To support the conclusion on PSII activity, I suggest determining the photochemical quenching coefficient of PS II Chl fluorescence (qP).
The radiogram band attribution figure 4 is only based on literature. I would recommend using western blot with commercially available antibodies to verify the band attribution. I do not understand why the authors did not perform this experiment with 18:3.
The regulation of D1 synthesis can be done at the transcriptional or translational levels. I would suggest to also check the mRNA level in presence of the different FFA to assess the level of regulation.
The authors are proposing that incubation with 18:3 makes, in fine, PSII unstable (line 189). This conclusion is in contradiction with line 89 (referring to figure2): “FFA has no effects on the stability of PSII”.
In the figure caption, it is stated that the “S.D. of results from three independent experiments”. Are those technical or biological replicates.
Minor comments:
Line 145: Panel (b) shows data for time=0 min, while these data are not present in the panel (a). I think this point has been extrapolated but not measured.
Lines 190-193: I do not understand these lines about α-tocopherol. Would you please reformulate them?
Lines 226-234: I do not know whether it is pertinent to control photosynthetic activity by adding exogenous FFA from an industrial point of view. As outlined (line 231), FFAs are “rapidly turned over by cellular metabolisms”, therefore the signalling molecule is very short-lived. Also, one should estimate whether the cost of such molecules is compatible with large-scale cultivation facilities. Industrial concerns are more about FFA production, and this research definitely shows that Lauric acid or alpha-linoleic acid hinders photosynthetic activity, while palmitic and stearic acids support it.
Line 239: Precise the light quality (white?)
Line 247: What is chlorophyll concentration?
Author Response
The authors proposed to use FFA to regulate photosynthetic activity (line 230) with the applied aim of improving the ability of photoprotection and biomass yield (line 227-228). Such as, the paper does not present any relevant data toward this aim. Therefore, I would recommend performing a growth curve in presence of the different FFA (added once or at several times when the FFA turned-over is fast (as written line 231)).
Response to the comment
Kojima et al. Appl. Microbiol. Biotech. (2016) (see also ref.13) has already reported that palmitic acid (16:0) or stearic acid (18:0) improves cellular growth, whereas lauric acid (12:0) or linolenic acid (18:3) inhibits it in Synechocystis sp. PCC 6803. We added this information to the revised manuscript (lines 70-71) and did not repeat the same experiments.
In cyanobacteria, photosynthesis and respiration occur in the same membrane. Lines 164-170, the authors stress out that in animals FFA enhanced ROS production by inhibiting the respiratory electron transport. This raises the question of whether this is not happening as well in cyanobacteria? Hindering the photosynthetic linear electron transport chain would result in a more reduced plastoquinone pool. Consequently, PSII damages will be enhanced. This could explain the additional photodamages observed in presence of α-linolenic acid (18:3). To rule out this possibility, I would suggest experiments such as P700 re-reduction kinetics. The oxygen evolution in the light is clearly impaired as shown in Figure 1. Since photosynthesis and respiration occur in the same membrane, both could be impaired. I would recommend verifying that the oxygen consumption in the dark is not changed in the presence of FFA.
Response to the comment
Since respiratory activity is very low under light conditions in Synechocystis, it is very likely that there is no enhancement of photoinhibition of PSII due to the respiratory electron transport activity. We agree with your point that excess ROS production by FFA has not been observed in Synechocystis. Thus, we deleted the discussion part about ROS in the revised manuscript.
To support the conclusion on PSII activity, I suggest determining the photochemical quenching coefficient of PS II Chl fluorescence (qP).
Response to the comment
As mentioned above, respiratory activity is very low under light conditions in Synechocystis. Thus, rate of oxygen evolution in the presence of artificial quinone under light conditions directly corresponds to PSII activity and the measurement of oxygen evolution is the most conventional method for the determination of PSII activity.
The radiogram band attribution figure 4 is only based on literature. I would recommend using western blot with commercially available antibodies to verify the band attribution. I do not understand why the authors did not perform this experiment with 18:3.
Response to the comment
The detection of the intermediate and premature D1 bands using 35S-Met and 35S-Cys is widely accepted in the field and the bands corresponding to D1 can be attributed without Western blot analysis. As we responded to the comment 3 by the reviewer 1, we are very interested in the effects of 18:3 on the de novo synthesis of D1 protein. But the mechanisms might be more complicated than those of other FFAs because 18:3 affects not only repair, but also photodamage to PSII. Thus, in the present study, we chose 12:0, 16:0, and 18:0 that affect only the repair of PSII to make the story simple and to avoid the confusion to the readers. We are planning to investigate the mechanisms how 18:3 affects the both repair and photodamage to PSII in the future.
The regulation of D1 synthesis can be done at the transcriptional or translational levels. I would suggest to also check the mRNA level in presence of the different FFA to assess the level of regulation.
Response to the comment
The turnover of D1 protein is regulated through a balance of processes between translation and degradation, but not at transcription in Synechocystis (Nishiyama et al. EMBO J. 2001) and Arabidopsis (Chotewutmontri and Barkan PNAS 2020). Thus, we did not check the mRNA level.
The authors are proposing that incubation with 18:3 makes, in fine, PSII unstable (line 189). This conclusion is in contradiction with line 89 (referring to figure2): “FFA has no effects on the stability of PSII”.
Response to the comment
Stability of PSII in line 89 means “thermostability,” whereas “unstable” refers to that the fact that 18:3 makes PSII unstable against light damage. We have then changed “stability” to “thermostability” in the lines 86 and 88 in the revised text.
In the figure caption, it is stated that the “S.D. of results from three independent experiments”. Are those technical or biological replicates.
Response to the comment
“Independent replicates” means biological replicates.
Minor comments:
Line 145: Panel (b) shows data for time = 0 min, while these data are not present in the panel (a). I think this point has been extrapolated but not measured.
Response to the comment
In this experiment, there is no radioactivity at time 0 as shown in Jimbo et al. Plant Physiol. 2019. Thus, it is not necessary to show the data at time 0 in panel (a) and radioactivity at time 0 in panel (b) should be zero.
Lines 190-193: I do not understand these lines about α-tocopherol. Would you please reformulate them?
Response to the comment
We agree with the reviewer that the sentences are a bit confusing. We edited this part in the revised text (lines177-181).
Lines 226-234: I do not know whether it is pertinent to control photosynthetic activity by adding exogenous FFA from an industrial point of view. As outlined (line 231), FFAs are “rapidly turned over by cellular metabolisms”, therefore the signalling molecule is very short-lived. Also, one should estimate whether the cost of such molecules is compatible with large-scale cultivation facilities. Industrial concerns are more about FFA production, and this research definitely shows that Lauric acid or alpha-linoleic acid hinders photosynthetic activity, while palmitic and stearic acids support it.
Response to the comment
Cyanobacteria are widely used as bioplatforms for bioenergy production. Since cyanobacteria do not have any degradative activity of FFA (beta-oxidation pathway), FFA could be used as a nice tool to control photosynthetic activity. Our future proposal described in section 3.3. is not only for bioenergy production, but also for the purification of environmental pollution by aquatic photosynthetic organisms. As mentioned in line 280 in the revised text, FFA have an advantage as it possesses a lighter environmental footprint in comparison to other chemicals.
Line 239: Precise the light quality (white?)
Response to the comment
White halogen light was used as the light source in all experiments (line 291).
Line 247: What is chlorophyll concentration?
Response to the comment
Chlorophyll concentration was added to the revised text in the line 288.
Round 2
Reviewer 4 Report
The authors convincingly answered my concerns and modified their text to provide the necessary information. They also remove the too far fetched interpretations and thus provide shorter but safer discussion.
In fact, the following point has not been addressed. The authors answered my concern about respiratory electron chain inhibition but not about the inhibition of the photosynthetic one: Hindering the photosynthetic linear electron transport chain would result in a more reduced plastoquinone pool. Consequently, PSII damages will be enhanced. This could explain the additional photodamages observed in presence of α-linolenic acid (18:3). To rule out this possibility, I would suggest experiments such as P700 re-reduction kinetics.
The new version of the discussion proposes (lines 185-190) that the delta pH across the membrane is altered, reducing the proton motive force (PMF) and enhancing PSII photodamages. I suggest to determine the electron transport rate (ETR) as performed in ref.22 (Ammonia Triggers Photodamage of Photosystem II in the Cyanobacterium Synechocystis sp. Strain PCC 6803 -